# Enzymatic Synthesis of New Acetoacetate–Ursodeoxycholic Acid Hybrids as Potential Therapeutic Agents and Useful Synthetic Scaffolds as Well

**DOI:** 10.3390/molecules29061305

**Published:** 2024-03-15

**Authors:** Valentina Venturi, Elena Marchesi, Daniela Perrone, Valentina Costa, Martina Catani, Simona Aprile, Lindomar Alberto Lerin, Federico Zappaterra, Pier Paolo Giovannini, Lorenzo Preti

**Affiliations:** 1Department of Environmental and Prevention Sciences, University of Ferrara, 44121 Ferrara, Italy; vntvnt@unife.it (V.V.); valentina.costa@unife.it (V.C.); prtlnz@unife.it (L.P.); 2Department of Chemical Pharmaceutical and Agricultural Sciences, University of Ferrara, 44121 Ferrara, Italy; mrclne@unife.it (E.M.); ctnmtn@unife.it (M.C.); simona.aprile@unife.it (S.A.); lrnldm@unife.it (L.A.L.); zppfrc@unife.it (F.Z.)

**Keywords:** biocatalysis, ursodeoxycholic acid, ketone bodies, acetoacetate, lipase, hybrid compounds

## Abstract

Ursodeoxycholic acid (UDCA) and acetoacetate are natural compounds present in the human intestine and blood, respectively. A number of studies highlighted that besides their well-known primary biological roles, both compounds possess the ability to influence a variety of cellular processes involved in the etiology of various diseases. These reasons suggested the potential of acetoacetate–UDCA hybrids as possible therapeutic agents and prompted us to develop a synthetic strategy to selectively derivatize the hydroxyl groups of the bile acid with acetoacetyl moieties. 3α-acetoacetoxy UDCA was obtained (60% isolated yield) via the regioselective transesterification of methyl acetoacetate with UDCA promoted by the *Candida antarctica* lipase B (CAL-B). 3α,7β-bis-acetoacetoxy UDCA was obtained instead by thermal condensation of methyl acetoacetate and UDCA (80% isolated yield). This bis-adduct was finally converted to the 7β-acetoacetoxy UDCA (82% isolated yield) via CAL-B catalyzed regioselective alcoholysis of the ester group on the 3α position. In order to demonstrate the value of the above new hybrids as UDCA-based scaffolds, 3α-acetoacetoxy UDCA was subjected to multicomponent Biginelli reaction with benzaldehyde and urea to obtain the corresponding 4-phenyl-3,4-dihydropyrimidin-2-(*1H*)-one derivative in 65% isolated yield.

## 1. Introduction

The therapy of complex diseases such as cancer, diabetes, neurodegenerative diseases, and heart failure often requires multi-drug therapy. Furthermore, the use of additional compounds able to deliver the active molecules toward specific organs and tissues may be needed to boost the efficacy of the pharmacological treatments. The use of hybrid compounds assembled by covalently binding two or more biologically active agents is an emerging strategy to pursue, especially when traditional combination therapy fails [1,2]. Molecular hybridization has been recognized as an efficient tool for overcoming several typical drug limitations since the covalent combination of two pharmacophores in a single unit could lead to a new hybrid molecule with an improved pharmacological potency compared to that of single drugs [1]. In this field, bile acids (BAs) and especially ursodeoxycholic acid (UDCA) play a relevant role either as the bioactive moiety or drug delivery scaffold [3]. UDCA is a primary BA in ursidae [4] and a secondary one in humans, where it is produced in the intestinal lumen by the bacterial-mediated epimerization of the 7α-hydroxyl group of the primary BA chenodeoxycholic acid (CDCA) [5]. In addition, used for the dissolution of cholesterol gallstones and the treatment of various liver diseases [6] (it is the only drug approved by the U.S. Food and Drug Administration for the treatment of primary biliary cirrhosis [7]), UDCA is now under investigation for its efficacy in the treatment of numerous conditions associated with inflammation and apoptosis including neurological [8] ocular [9], bowel [5,10], and cardiovascular diseases [11,12,13]. In recent years, the anticancer potential of UDCA has been highlighted by several studies evidencing its capability to limit tumor cell growth and to modulate different molecular pathways implicated in tumor cell growth and/or cell death [14]. Furthermore, UDCA’s ability to reduce susceptibility to SARS-CoV-2 has been demonstrated in ex vivo experiments, and the effectiveness of UDCA for the prevention of SARS-CoV-2 infection is currently under clinical investigation [15]. Likewise, to BAs, the ketone bodies (KBs) acetoacetate (AcAc) and β-hydroxybutyrate (BHB) are human endogenous compounds that, in addition to their main role as energy substrates, show properties in regulating cellular processes involved in various pathological conditions. The primary KB AcAc is produced by hepatocyte mitochondria from fatty acid-derived acetyl-coenzyme A. The equilibrium between AcAc and its reduced counterpart BHB is influenced by the NAD^+^/NADH ratio and is mediated by the mitochondrial BHB dehydrogenase 1 [16]. Under hypoglycemic conditions, fatty acid oxidation increases and determines an enhancement of the KB blood concentration. During this metabolic state, known as ketosis, the circulating KBs become the main metabolic fuel. In addition to this primary metabolic role, KBs have been shown to influence a variety of cellular processes, including gene transcription, inflammation, and oxidative stress [17,18]. These mechanisms seem to be responsible for the beneficial effects of ketosis on a wide array of pathologies, such as neurodegenerative diseases, heart failure, and cancer. From what is reported above, it emerges that UDCA and KBs have overlapping potential therapeutic areas. Furthermore, they also have the common ability to cross the blood–brain barrier [8,19]. Quite curiously, during our recent research focused on KB activities as well as on the synthesis of KB derivatives, we found very little information about the combined activities of UDCA and KBs [20] and, in spite of the large number of BA conjugates reported in the literature [21,22,23,24,25,26], no article has reported the synthesis of covalent UDCA-KB conjugates (this is the same for the other BAs). In order to close this gap, in this work, we describe the enzymatic synthesis of new acetoacetic esters of UDCA. The simple synthesis and purification herein reported allow us to obtain three new hybrid compounds on a gram scale. The novel compounds are fully characterized and represent potential pro-drugs whose activity can be assessed in the future by in vitro and/or in vivo studies. Furthermore, thanks to the reactivity of the added acetoacetyl moieties, the above hybrids constitute innovative scaffolds suited for further chemical elaboration. To demonstrate this, we herein also report the conversion of the 3α-acetoacetoxy UDCA into the corresponding 4-phenyl-3,4-dihydropyrimidin-2-(*1H*)-one derivative via the multicomponent Biginelli reaction with urea and benzaldehyde.

## 2. Results and Discussion

### 2.1. State of the Art on Enzyme-Catalyzed Esterification of Bile Acids

The enzymatic acylation of the hydroxyl groups of bile acids has been explored by several groups. The regioselective synthesis of the 3α-acetoxy derivatives of the lithocolic, chenodeoxycholic, deoxycholic, and hyocholic acids has been recently reported by Baldessari and coworkers [27,28], who employed the *Candida antractica* lipase B (CAL-B) and ethyl acetate as the catalyst and the acylating agent, respectively. The same authors reported that this approach failed with cholic acid, which needs to be converted into its ethyl ester to be successfully acetylated to the corresponding 3α-acetoxy derivative, confirming the narrow regioselectivity of CAL-B for the position 3 [29]. The methyl ester of cholic acid was instead employed as a substrate in acylation reactions performed with long-chain fatty acid methyl esters as the acylating agents and CAL-B as the catalyst, also obtaining, in this case, the corresponding 3α-acyloxy derivatives [30]. Similar results have been reported for the transesterification of methyl butanoate with the methyl esters of the cholic, deoxycholic, chenodeoxycholic, and ursodeoxycholic acids promoted by the lipase from *Candida cylindracea* (Type VII) [31]. In spite of the therapeutic relevance of UDCA, to the best of our knowledge, this last study is the only one which reports the enzymatic acylation of this bile acid (although under the form of methyl ester). The literature overview resumed in Table 1 also highlighted that, although various acetoacetate esters have been already produced via enzymatic transesterification [32], the enzymatic synthesis of acetoacetic esters of bile acids is unprecedented. In fact, the preparation of acetoacetoxy derivatives of several bile acids (not including UDCA) has only been described in two patents where the transesterification reactions were conducted by exploiting acidic [33] or thermal catalysis [34].

### 2.2. Screening of the Biocatalysts

This evidence prompted us to explore the activity of six lipase-based biocatalysts on the transesterification of methyl acetoacetate with UDCA (Table 2, compounds **2** and **1**, respectively). We decided to employ the free bile acid instead of its methyl or ethyl ester to limit the changes in the natural structure of the biologically active moieties fused in the resulting hybrids. The screening was conducted by adding 500 U of the enzyme to a solution of **1** (0.25 g) and **2** (5 equivalents) in *t*-butanol (5 mL). The reactions were shaken at 50 °C for 24 h, and after removing the enzyme, solvent, and excess of methyl acetoacetate, the residues were analyzed by TLC and ^1^H-NMR analyses. As reported in Table 2, only the reactions performed in the presence of the biocatalysts Lipozyme 435 and Lipura Flex, both containing *C. antarctica* lipase B as the enzyme, showed the presence of new products in addition to the starting material. The ^1^H-NMR spectra of both the crude residues showed the presence of a multiplet between 4.79 and 4.69 ppm that, by comparison with the literature data [35], we attributed to the resonance of the C3 proton after the esterification of the corresponding hydroxyl group. The additional presence of two singlets centered at 3.41 and 2.25 ppm attributable to the methylene and methyl groups of the acetoacetyl moiety confirmed the formation of the 3α-acetoacetoxy derivative **3a** (Table 2). By ^1^H NMR analysis, we calculated conversions of 57 and 61% for the reactions performed with Lipozyme 435 and Lipura Flex, respectively (see Appendix A). The product **3a** was purified by column chromatography on silica gel and fully characterized.

### 2.3. Optimized Synthesis of the 3α-Acetoacetoxy UDCA ***3a***

Once the CAL-B Lipura Flex was identified as the suitable biocatalyst, the effect of the substrate’s molar ratio was investigated by repeating the reaction using 2.5, 5, and 10 equivalents of acetoacetate **2** and monitoring the conversion by ^1^H-NMR analysis (Figure 1A). The best result (60%) was obtained after 24 h at 50 °C with five equivalents of **2**. A slight drop in the conversion was observed at longer reaction times (56% after 48 h). The reaction conducted with 2.5 equivalents of **2** instead reached a maximum conversion of 36% after 32 h, confirming a small decrease in the conversion at longer reaction times (34% after 48 h). Quite surprisingly, increasing the excess of **2** from 5 to 10 equivalents, we observed a marked decrease in the maximum conversion, which reached the higher value of 26% after 48 h. Based on these results, we engaged the study on the effect of the temperature by repeating the reaction with five equivalents of **2** at 40 and 70 °C (Figure 1B). As predicted, the reaction performed at 40 °C showed a slower reaction rate, reaching the equilibria after 24 h but with a lower yield of 41%. Also, the reaction performed at 70 °C afforded a worse maximum yield (51% after 24 h) with respect to that conducted at 50 °C, showing, in addition, a more marked decrease at longer reaction times. Under the optimized conditions of 50 °C and five equivalents of **2**, the biocatalyst reuse was evaluated. Six reactions were performed consequently with the same biocatalyst; the yields were comparable for the first five reaction cycles, while the sixth one showed a 20% decrease in the yield (see Appendix A).

### 2.4. Chemo-Enzymatic Synthesis of the 3α,7β-bis-Acetoacetoxy UDCA ***3b*** and 7β-Acetoacetoxy UDCA ***3c***

Once we optimized the reaction parameters for the enzymatic synthesis of **3a**, we developed a chemo-enzymatic strategy to obtain the other two theoretical products of the transesterification of the methyl acetoacetate **2** with UDCA **1**, namely the 3α,7β-*bis*-acetoacetoxy UDCA **3b** (Figure 1) and the 7β-acetoacetoxy UDCA **3c** (Table 3). Inspired by the patent of Cummings al. [34], we explored the thermal condensation of substrates **1** and **2** as the strategy to obtain **3b**. Substrate **1** was dissolved into an excess of **2** (20 equivalents), and the mixture was warmed to 120 °C. After 2 h, the TLC analysis showed complete disappearance of the limiting reagent **1** with the formation of a major product accompanied by trace amounts of side products, identified as the monoacylated intermediates **3a** and **3c**. The mixture was evaporated to remove the excess of **2** and chromatographed to obtain the pure product **3b** (80% yield). The absence of any signal in the region between 3.6 and 3.5 ppm and the appearance of two multiplets at 4.85–4.77 and 4.77–4.68 ppm confirmed the esterification of both the C7 and C3 hydroxyl groups, respectively. Additional signals that contributed to confirm the structure of **3b** were those due to the resonance of the methylene (3.41 and 3.37 ppm) and methyl groups (2.26 and 2.25 ppm) of the acetoacetyl moieties.

With the compound **3b** in hand, we attempted to produce the 7β-acetoacetoxy derivative **3c** via the selective enzymatic alcoholysis of **3b** (Table 3) by following the approach proposed by Baldessari et al. [27,28,29]. The reaction was performed by adding the CAL-B to a solution of **3b** and ethanol (five equivalents) in toluene. The reaction was gently shaken at 50 °C and monitored by TLC. After 6 h, most of the substrate was converted to the expected product **3c**, but pushing the conversion to completion by leaving the reaction running for a further 6 h, an additional compound appeared. Hence, after removing the enzyme and the solvent, the residue was chromatographed, obtaining the pure product **3c** (50% yield) and its ethyl ester derivative **3d** (33% yield). Considering this result, the reaction was repeated in the presence of two equivalents of ethanol. By running the reaction for 8 h under this condition, the desired product **3c** was obtained in 82% isolated yield.

### 2.5. Synthesis of the 3α-(4-Phenyl-6-methyl-2-oxo-1,2,3,4-tetrahydropyrimidine-5-carboxyl)-UDCA ***4a***

The three new products, **3a**, **3b,** and **3c**, not only represent hybrids of the two bioactive parent compounds UDCA and acetoacetate but can also be considered useful scaffolds for the synthesis of further UDCA derivatives. In fact, the added β-keto ester moieties constitute valuable reactive structures that can be converted to a wide variety of molecular systems thanks to the presence of two different electrophilic carbonyls and two nucleophilic carbons, which can react selectively under suitable conditions [36]. Just to give a demonstration of this, we employed product **3a** for the synthesis of the dihydropyrimidinone derivative **4a** by exploiting the multicomponent Biginelli cyclocondensation with benzaldehyde and urea catalyzed by Yb(OTf)_3_ (Figure 2). An equimolar mixture of **3a** and benzaldehyde in dry THF was added with three equivalents of urea and 50% mol of Yb(OTf)_3_. After 24 h at 70 °C, the reaction was treated as described in the Section 3.5 and chromatographed on silica gel to afford the expected product **4a** in 70% yield as a 1:1 diastereomeric mixture determined from the ratio of the integrals relative to the H4 proton of the heterocycle. Analytical samples of the two diastereoisomers were separated by preparative TLC and characterized by ^1^H and ^13^C NMR analyses.

## 3. Materials and Methods

### 3.1. General Information

The biocatalysts Lipozyme 435, Novocor AD L, Lipura Flex, Lipozyme RM IM, Lipozyme TL IM, and Palatase 20000 L were kindly provided by Novozymes (Lyngby, Denmark). Methyl acetoacetate and ursodeoxycholic acid are commercially available. Reactions were monitored by TLC on silica gel 60 F254 with detection by charring with phosphomolybdic acid. Flash column chromatography was performed on silica gel 60 (230–400 mesh). ^1^H (400 MHz) and ^13^C (101 MHz) NMR spectra were recorded in CDCl_3_ solutions at room temperature unless otherwise stated. Chemical shifts (δ) were reported in ppm relative to residual solvent signals. Peak assignments were aided by ^1^H–^1^H COSY and gradient-HMQC experiments. Optical rotations were measured at 20 ± 2 °C in the solvents specified below, and the [α]^20^_D_ values are given in 10^−1^ deg cm^2^ g^−1^. The mass of synthesized compounds was assessed by injecting 1 µL of each sample into a Vanquish Flex Ultra High-Performance Liquid Chromatography (UHPLC) system coupled to a High-Resolution Orbitrap Exploris 240 mass spectrometer ThermoFisher Scientific, (Waltham, MA USA). The separation was performed on a Waters BEH C18 column (100 × 2.1 mm L × I.D., 1.7 µm) operated under reversed phase conditions by using water and acetonitrile + 0.1% formic acid as mobile phase. All the samples, except from compound **3d**, were analyzed in negative mode. The results are, in most of the cases, below ±0.4 ppm, except for compound **3d** (±5 ppm).

### 3.2. Optimized Procedure for the Synthesis of 3α-Acetoacetoxy Ursodeoxycholic Acid ***3a***

Ursodeoxycholic acid **1** (250 mg, 0.634 mmol) and methyl acetoacetate **2** (370 mg, 3.19 mmol) were dissolved in *t*-butanol (5.0 mL). Lipuraflex (50 mg, 500 U) was added, and the mixture was shaken at 50 °C for 24 h. The biocatalyst was removed by filtration, and the filtrate was evaporated under reduced pressure to remove the solvent and the excess of **2**. The residue was chromatographed on silica gel using cyclohexane/ethyl acetate 1.5:1, containing 2% acetic acid as the eluent. The pure product **3a** (182 mg, 0.38 mmol) was obtained in 60% yield. [α]^D^_20_ = +53.3 (c 2.4, CDCl_3_). ^1^H-NMR (400 MHz, CDCl_3_) δ = 4.79–4.69 (m, 1H, H-3α), 3.62–3.52 (m, 1H, H-7β), 3.41 (s, 2H, OC-CH_2_-CO), 2.45–2.21 (m, 2H, H-23), 2.25 (s, 3H, CH_3_-CO), 2.04–1.02 (m, 24H), 0.95 (s, 3H, H-18), 0.94 (d, 3H, *J* = 6.3 Hz, H-21), 0.68 (s, 3H, H-19). ^13^C-NMR (101 MHz, CDCl_3_) δ = 200.76, 179.62, 166.61, 75.03, 71.24, 55.65, 54.91, 50.47, 43.73, 43.63, 42.24, 40.04, 39.16, 36.52, 35.18, 34.51, 34.06, 32.93, 30.97, 30.80, 30.11, 28.56, 26.80, 26.33, 23.30, 21.19, 18.35, 12.12. HRMS (ESI) *m*/*z* calcd for C_28_H_43_O_6_^−^: 475.3065 [M − H]^−^; found: 475.3063.

### 3.3. Procedure for the Synthesis of 3α,7β-bis-Acetoacetoxy Ursodeoxycholic Acid ***3b***

Ursodeoxycholic acid **1** (250 mg, 0.634 mmol) was dissolved in methyl acetoacetate **2** (1.37 mL, 12.7 mmol), and the mixture was warmed at 120 °C for 2 h. After cooling, the residue was evaporated to remove the excess of **2,** and the residue was chromatographed on silica gel using cyclohexane/ethyl acetate 1.5:1, containing 2% acetic acid as the eluent. Pure product **3b** (285 mg, 0.0.51 mmol) was obtained in 80% yield. [α]^D^_20_ = +30 (c 3.0, CDCl_3_). ^1^H-NMR (400 MHz, CDCl_3_) δ = 4.85–4.77 (m, 1H, H-7β), 4.77–4.68 (m, 1H, H-3α), 3.40 (s, 2H, OC-CH_2_-CO), 3.37 (d, 2H, *J* = 1.9 Hz, OC-CH_2_-CO), 2.43–2.20 (m, 2H, H-23), 2.46 (s, 3H, CH_3_-CO), 2.24 (s, 3H, CH_3_-CO), 2.03–1.01 (m, H-24), 0.96 (s, 3H, H-18), 0.92 (d, 3H, *J* = 6.3 Hz, H-21). 0.67 (s, 3H, H-19).^13^C NMR (101 MHz, CDCl_3_) δ = 200.62, 200.52, 179.41, 166.68, 166.57, 74.95, 74.69, 55.14, 54.94, 50.68, 50.42, 43.57, 42.00, 39.80, 39.38, 35.14, 34.34, 33.95, 32.60, 30.82, 30.68, 30.18, 30.14, 28.33, 26.26, 25.70, 23.18, 21.18, 18.31, 12.06. HRMS (ESI) *m*/*z* calcd for C_32_H_47_O_8_^−^: 559.3276 [M − H]^−^; found: 559.3273.

### 3.4. Procedure for the Synthesis of 7β-Acetoacetoxy Ursodeoxycholic Acid ***3c*** and of the Corresponding Ethyl Ester ***3d***

Lipuraflex (50 mg, 500 U) was added to a solution of **3b** (400 mg, 0.71 mmol) and ethanol (82 mL, 1.42 mmol) in toluene (20 mL), and the mixture was shaken at 50 °C for 8 h. After that, the biocatalyst was removed by filtration, the filtrate was evaporated, and the residue was chromatographed on silica gel using cyclohexane/ethyl acetate 1.5:1 containing 2% of acetic acid as the eluent. The pure product **3c** (277 mg, 0.58 mmol) was obtained in 82% yield. [α]^D^_20_ = +24 (c 3.5, CDCl_3_). ^1^H-NMR (400 MHz, CDCl_3_) δ = 4.86–4.77 (m, 1H, H-7β), 3.64–3.52 (m, 1H, H-3α), 3.37 (s, 2H, CH_2_-CO), 2.47–2.18 (m, 2H, H-23), 2.45 (s, 3H, CH_3_-CO), 2.02–0.97 (m, 24H), 0.94 (s, 3H, H-18), 0.91 (d, 3H, *J* = 6.3 Hz, H-21), 0.66 (s, 3H, H-19). ^13^C NMR (101 MHz, CDCl_3_) δ = 200.73, 179.82, 166.71, 75.28, 71.30, 55.14, 54.93, 50.70, 43.57, 42.15, 39.86, 39.83, 39.35, 36.89, 35.16, 34.72, 33.91, 32.77, 30.99, 30.70, 30.20, 30.02, 28.34, 25.73, 23.21, 21.14, 18.31, 12.05. HRMS (ESI) *m*/*z* calcd for C_28_H_43_O_6_^−^: 475,3065 [M − H]^−^; found: 475.3063. The reaction was performed with 5 equivalents of ethanol (0.2 mL) for 12 h after chromatography afforded a mixture of the products **3c** (50%) and **3d** (33%). 7β-acetoacetoxy ethyl ursodeoxicholate **3d**: [α]^D^_20_ = +31 (c 2.8, CDCl_3_). ^1^H-NMR (400 MHz, CDCl_3_) δ = 4.86–4.78 (m, 1H, H-7β), 4.11 (q, 2H, *J* = 7.1 Hz, CH_2_-CH_3_), 3.63–3.53 (m, 1H, H-3α), 3.36 (d, 2H, *J* = 1.76 Hz, OC-CH_2_-CO), 2.37–2.14 (m, 2H, H-23), 2.49 (s, 3H, CH_3_-CO), 2.01–0.88 (m, 24H), 1.24 (t, 3H, J = 7.1 Hz, CH_3_-CH_2_), 0.95 (s, 3H, H-18), 0.91 (d, 3H, *J* = 6.4 Hz, H-21), 0.65 (s, 3H, H-19). ^13^C NMR (101 MHz, CDCl_3_) δ = 200.63, 174.25, 166.66, 75.25, 71.26, 60.19, 55.17, 54.94, 50.71, 43.56, 42.17, 39.87, 39.85, 39.37, 37.03, 35.17, 34.73, 33.92, 32.79, 31.29, 30.93, 30.19, 30.12, 28.34, 25.74, 23.22, 21.14, 18.34, 14.25, 12.03. HRMS (ESI) *m*/*z* calcd for C_30_H_49_O_6_^+^: 505.3529 [M + H]^+^; found: 505.3493.

### 3.5. Procedure for the Synthesis of 3α-(4-Phenyl-6-methyl-2-oxo-1,2,3,4-tetrahydropyrimidine-5-carboxyl)-UDCA ***4a***

Compound **3a** (100 mg, 0.21 mmol), urea (38 mg, 0.63 mmol), and Yb(TfO)_3_ (65 mg, 0.1 mmol) were dissolved in 0.5 mL of freshly distilled THF in the presence of 4 Å molecular sieves (20 mg). To this mixture, 21 µL (0.21 mmol) of benzaldehyde was added, and the mixture was stirred at 70 °C for 24 h. The reaction mixture was then diluted with EtOAc (20 mL) and filtered through Celite^®^. The organic solvent was washed with H_2_O (2 × 10 mL), dried over anhydrous sodium sulfate, and evaporated under a vacuum. The residue was purified by flash chromatography using EtOAc/acetic acid 1:1 as the eluent to yield product **4a** (83 mg, 65%) as a diastereomeric mixture of approximately 5:1 ratio, as calculated by ^1^H-NMR analysis. Analytical samples of the two diastereoisomers were obtained by preparative TLC using the above eluent. Diastereomer A (less polar): [α]^D^_20_ = −11.2 (c 0.5, CDCl_3_). ^1^H NMR selected data (400 MHz, CDCl_3_) δ = 7.50 (s, 1H, NH), 7.41–7.28 (m, 5H, Ph), 6.63 (s, 1H, NH), 5.45 (d, 1H, *J* = 3.0 Hz, CH-Ph DHMP), 4.76–4.47 (m, 1H, H-3α), 3.56–3.40 (m, 1H, H-7β), 2.36 (s, 3H, CH_3_ DHMP), 0.95 (d, *J* = 6.3 Hz, 3H, H-21), 0.92 (s, 3H, H-19), 0.69 (s, 3H, H-18). Diastereomer B (most polar): [α]^D^_20_ = +11.5 (c 0.5, CDCl_3_). ^1^H NMR selected data (400 MHz, CDC_3_) δ = 7.58 (s, 1H, NH), 7.35–7.26 (m, 5H, Ph), 6.63 (s, 1H, NH), 5.44 (d, 1H, *J* = 3.1 Hz, CH-Ph DHMP), 4.81–4.52 (m, 1H, H-3α), 3.64–3.53 (m, 1H, H-7β), 2.36 (s, 3H, CH_3_ DHMP), 0.97–0.91 (m, 6H, H-21 and H-19), 0.69 (s, 3H, H-18). HRMS (ESI) *m*/*z* calcd for C_36_H_49_N_2_O_6_^−^: 605.3596 [M − H]^−^; found: 605.3596.

## 4. Conclusions

The herein-reported study meets the emerging demand for new hybrid compounds assembled by covalently binding two or more biologically active agents. Recent studies have highlighted that ursodeoxycholic acid and the ketone body acetoacetate are involved in the regulation of numerous cellular processes related to a wide spectrum of multifactorial diseases, such as neurological, cardiovascular, and cancer. Starting from this evidence, we successfully engaged the first synthesis of the unprecedented covalent UDCA–acetoacetate conjugates 3α-acetoacetoxy UDCA **3a**, 3α,7β-*bis*-acetoacetoxy UDCA **3b,** and 7β-acetoacetoxy UDCA **3c**. These three new compounds were selectively obtained with high purity and satisfactory yields via a simple chemo-enzymatic approach, which exploits the selectivity of the lipase B from *Candida antarctica*. The new products were fully characterized and, thanks to the reactivity of the added acetoacetate ester moieties, can also be considered useful UDCA-based scaffolds for the preparation of further derivatives. As proof of this concept, the 3α-acetoacetoxy UDCA **3a** was converted into the corresponding 4-phenyl-3,4-dihydropyrimidin-2-(1*H*)-one **4a** derivative exploiting the multicomponent Biginelli reaction.

## Data Availability

Data are contained within the article and Appendix A.

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
