# Peer review of "Enzymatic Synthesis of New Acetoacetate–Ursodeoxycholic Acid Hybrids as Potential Therapeutic Agents and Useful Synthetic Scaffolds as Well"

_molecules, 2024, doi:10.3390/molecules29061305_

Round 1
Reviewer 1 Report
Comments and Suggestions for Authors
The authors accomplished chem-enzymatic synthesis of three types of ursodeoxycholic acid derivatives mainly through lipase-catalyzed regioselective transesterification. The work has been carefully investigated and the manuscript is written well. Therefore, this reviewer feels that this work is worth for publication on this journal after minor revision. My concerns are as follows.
(1) Lipase-catalyzed synthesis of acetoacetate ester was reported by Cordova and Janda. I strongly recommend that the author cite their paper and mentioned; Armando Córdova and Kim D. Janda, A Highly Chemo- and Stereoselective Synthesis of b-Keto Esters via a Polymer-Supported Lipase Catalyzed Transesterfication, J. Org. Chem. 2001, 66, 1906-1909, doi:10.1021/jo001478o. According to this, the reference numbering system should be revised.
(2) Page 1, line 21: “Candida antarctica” should be typed by italic fonts.
(3) Page 5, Figure 1. Line 168. “1H” should be typed as “1H”.
(4) Page 6, Table 3: “1 to 2 molar ratio” is wrong. This should be revised as “Molar ratio of 3b to ethanol”.
Author Response
The revisions highlighted by the Referee 1 have been fully accepted and the manuscript has been revised as follow.
The article by A. Córdova and K. D. Janda, (J. Org. Chem. 2001, 66, 1906-1909), has been cited with the reference number 32 in the revised version of the manuscript. The sentence from line 104 to line 107 has been modified to better introduce the citation.
The name “Candida antarctica” at page 1, line 21, has been typed in italic font.
The proton’s symbol in the footnote of Figure 1 at page 5 has been corrected from “1H” to “1H”.
The header of the first column of Table 3 at page 6 has been corrected from “1 to 2 molar ratio” to “Molar ratio of 3b to ethanol”.
Reviewer 2 Report
Comments and Suggestions for Authors
In this work the authors describe the enzymatic synthesis of three new acetoacetic esters of ursodeoxycholic acid (UDCA-acetoacetate conjugates 3a-acetoacetoxy UDCA 3a, 3a,7b-bis-acetoacetoxy UDCA 3b, and 7b-acetoacetoxy UDCA 3c). The novel compounds have been fully characterized and they could be represent potential pro-drugs which activity in vitro and/or in vivo studies. Furthermore, the above hybrids constitute innovative scaffolds suited for further chemical elaboration. To demonstrate this, the authors herein also report the conversion of the 3a-acetoacetoxy UDCA 3a into the corresponding 4-phenyl-3,4-dihydropyrimidin2-(1H)-one derivative 4a through the multicomponent Biginelli reaction with urea and benzaldehyde.
In general terms the work is well developed. However, I consider that the authors should clarify and complement the following information:
1. In the case of the formation of the diastereomers in the Biginelli reaction for substrate 3a, the authors mention in line 224 that they were obtained in a 1:1 ratio, but in line 314 they mention that they were obtained in a 5:1 ratio. Please clarify this situation.
2. They also mention that they were able to separate each diastereomer by preparative TLC. If so, what are the optical rotation values and also if you were able to assign the absolute settings for each of them.
Once these questions are clarified, I consider that it can be accepted for publication.
Author Response
The revisions of the Referee 2 have been accepted by modifying the manuscript as below reported.
The diastereomeric ratio of the compound 4a is 1:1 as correctly reported in line 224 and documented by the chromatogram in the Supporting Information (S12). In virtue of this, the wrong value reported in line 314 (5:1) has been corrected.
By repeating several preparative TLC separations, we obtained amounts of the two epimers of the compound 4a suited for the measuring of the optical rotation values. These resulted -11.2 (c 0.5, CDCl3) for the first eluted one and +11.5 (c 0.5, CDCl3) for the second eluted one. These values have been added into the paragraph 3.5. of the Matherials Methods section (lines 317 and 321). Regarding to the determining of the absolute stereochemistry, we should have bigger amounts of the pure epimers to perform the hydrolysis followed by purification of the 4-phenyl-6-methyl-3,4-dihydropyrimidin-2-(1H)-one-5-carboxylic acid released in order to perform CD analyses according to Uray, Georg; Verdino, Petra; Belaj, Ferdinand; Kappe, C. Oliver; Fabian, Walter M. F. Journal of Organic Chemistry (2001), 66(20), 6685-6694. In our opinion this study falls out from the purpose of this work.